# De Novo Complement-Binding Anti-HLA Antibodies in Heart Transplanted Patients Is Associated with Severe Cardiac Allograft Vasculopathy and Poor Long-Term Survival

**DOI:** 10.3390/jcm11133731

**Published:** 2022-06-28

**Authors:** Guillaume Baudry, Matteo Pozzi, Matthieu Aubry, Elisabeth Hugon-Vallet, Raluca Mocan, Lara Chalabreysse, Philippe Portran, Jean-François Obadia, Olivier Thaunat, Nicolas Girerd, Valérie Dubois, Laurent Sebbag

**Affiliations:** 1Heart Failure and Transplant Department, Hospices Civils de Lyon, Louis Pradel Hospital, 69500 Bron, France; matthieu.aubry@chu-lyon.fr (M.A.); elisabeth.hugon-vallet@chu-lyon.fr (E.H.-V.); ralucapisau@yahoo.com (R.M.); laurent.sebbag@chu-lyon.fr (L.S.); 2Centre d’Investigations Cliniques Plurithématique 1433, Université de Lorraine, INSERM DCAC, CHRU de Nancy, F-CRIN INI-CRCT, 54500 Vandœuvre-lès-Nancy, France; n.girerd@chru-nancy.fr; 3Department of Cardiovascular and Thoracic Surgery, Hospices Civils de Lyon, 69500 Bron, France; matteo.pozzi@chu-lyon.fr (M.P.); jean-francois.obadia@chu-lyon.fr (J.-F.O.); 4Department of Pathology, Groupement Hospitalier Est, Hospices Civils de Lyon, 69500 Bron, France; lara.chalabreysse@chu-lyon.fr; 5Department of Anesthesiology and Critical Care, Hôpital Cardiologique Louis Pradel, 69500 Bron, France; philippe.portran@chu-lyon.fr; 6Department of Transplantation, Nephrology and Clinical Immunology, Hospices Civils de Lyon, 69003 Lyon, France; olivier.thaunat@chu-lyon.fr; 7EFS Auvergne Rhône Alpes, Laboratoire HLA, 111 rue Elisée Reclus, 69150 Décines, France; valerie.dubois@efs.sante.fr

**Keywords:** heart transplant, immunology, de novo DSA, complement-binding DSA, cardiac allograft vasculopathy

## Abstract

Introduction: De novo anti-HLA donor specific antibodies (DSA) have been inconsistently associated with cardiac allograft vasculopathy (CAV) and long-term mortality. We tested whether C3d-binding de novo DSA were associated with CAV or long-term-survival. Methods: We included 282 consecutive patients without preformed DSA on coronary angiography between 2010 and 2012. Angiographies were classified according to CAV ISHLT grading. The primary outcome was a composite criterion of severe CAV or mortality. As the impact of de novo antibodies should be assessed only after appearance, we used a Cox regression with time-dependent covariables. Results: Of the 282 patients, 51(18%) developed de novo DSA during follow-up, 29 patients had DSA with C3d-binding ability (DSA+C3d+), and 22 were without C3d-binding ability (DSA+C3d-). Compared with patients without DSA, DSA+C3d+ patients had an increased risk for the primary outcome of severe CAV or mortality (adjusted HR = 4.31 (2.40–7.74) *p* < 0.001) and long-term mortality (adjusted HR = 3.48 (1.97–6.15) *p* < 0.001) whereas DSA+C3d- did not (adjusted HR = 1.04 (0.43–2.47) *p* = 0.937 for primary outcome and HR = 1.08 (0.45–2.61) *p* = 0.866 for mortality). Conclusion: According to this large monocentric study in heart transplant patients, donor specific antibodies were associated with worse clinical outcome when binding complement. DSA and their complement-binding ability should thus be screened for to optimize heart transplant patient follow-up.

## 1. Introduction

Despite the good overall results of solid organ transplant and the improvement over time, the impact of anti-HLA antibodies on function and survival of heart allograft remains an issue in the management of patients [1,2,3,4,5]. Although proof regarding the interest in pre-transplant antibody detection and post-transplant monitoring of donor-specific anti-HLA antibodies (DSA) have been accumulating, a recent ISHLT consensus document has pointed out the heterogeneity of practice [6]. This heterogeneity highlights the complex role of donor-specific anti-HLA antibodies in antibody-mediated rejection (AMR) and late graft loss. The role of antibodies in late kidney graft failure and AMR has emerged in the past decade [7,8,9]. More recently, authors report deleterious impact of de novo donor-specific anti-HLA antibodies compared with preformed donor-specific anti-HLA antibodies [10,11,12]. However, the impact of DSA on rejection and allograft survival is heterogenous. The ability of binding complement is associated in kidney transplantation with worse outcome [13,14,15]. Indeed, it increases the rate of antibody-mediated rejection and severe graft injury as assessed by complement fraction C4d deposition within graft capillaries [16]. C3d-binding DSA were independent predictors for renal graft loss [17]. In cardiac transplantation, circulating DSA are associated with C4d deposition and cardiac allograft vasculopathy (CAV) [18,19] In the pediatric heart transplant population, complement-binding DSA was associated with a risk of graft loss and CAV [20]. In the adult population of heart transplants, Zhang et al. have shown the impact of DSA (preformed and de novo) on AMR [21]. This risk increased with the ability of binding C3d and was associated with complement deposition on the graft [21]. However, the long-term impact of complement-binding donor-specific anti-HLA antibodies in long-term adults and heart transplant recipient survivors is still a subject of debate. We sought to examine the association between C3d binding DSA and outcomes in our population of cardiac allograft recipients.

## 2. Materials and Methods

### 2.1. Patients’ Selection

This study was a monocentric cohort analysis. All heart transplanted patients with at least one coronary angiography performed in routine follow-up between 2010 and 2012 were included. A serum sampled to assess HLA antibody was drawn at the same time. Exclusion criteria was the presence of anti-HLA antibodies at the time of transplantation, coronary angiography performed outside routine protocol (severe graft dysfunction or coronary syndrome), and absence of HLA blood sampled at time of coronary angiography, 287 patients met the inclusion criteria. They had received a first HT between 1981 and 2012 and were negative for anti-HLA antibodies at the time of transplantation. Regarding the anti-HLA antibody status, 231 patients were found negative and 51 patients were positive; 5 patients were found positive for non-donor-specific anti-HLA antibodies. Patients only positive for non-donor-specific anti-HLA antibodies (n = 5) were excluded from further analysis. 

### 2.2. Cardiac Allograft Vasculopathy Evaluation

Our center’s protocol relies on a coronary angiography performed at 1-, 5-, and 10-years post-transplant and then every 2 years. Coronary angiographies were reloaded and scored blindly regarding patient DSA status by senior cardiologist according to ISHLT CAV scoring recommendations [22]. Severe CAV was defined by an ISHLT CAV score of 2 or more [22]. 

### 2.3. HLA Antibody Screening during Follow Up

Donors and recipients were typed for HLA class I (HLA-A and B) and class II (HLA-DR and DQ) locus either by serology (complement-dependent cytotoxicity assay) or by molecular biology such as reverse PCR-SSO assay using Luminex beads (Labtype^®^ SSO, One Lambda, Canoga Park, CA, USA) according to the manufacturer’s protocol.

Blood samples had been obtained annually in accordance with our protocol for all patients during the regular follow-up performed after transplantation and frozen sera had been stored. Post-transplantation monitoring for DSA were performed at 1, 3, 6, and 12 months post-operatively according to ISHLT guidelines [6]. Beyond one year post transplant, patients without DSA were monitored annually or more frequently in case of sensitization. Patients with de novo anti-HLA antibodies detected with the sensitive Luminex Single Antigen assay were considered positive. For these positive patients, previously stored samples were then checked to ascertain the outcome of sensitization occurrence. When no anti-HLA antibodies were detected at the time of or later than coronary angiography patients were considered negative for this analysis.

### 2.4. Methods for Assessing the Presence of DSA and Complement Components Binding Ability

Anti-HLA antibody screening was performed using Luminex screening assay (Lifecodes Lifescreen Deluxe, Immucor, Stamford, CT, USA). Positivity of the evaluated samples was assessed using the manufacturer’s cutoff. When the screening test was found positive, a specific identification was performed with a set of 96 (HLA class I) and 96 (HLA class II) color-coded beads coated with recombinant HLA antigens (Lifecodes Single Antigen (LSA), Immucor, Stamford, CT, USA) according to the manufacturer’s protocol. Then, for all patients with positive DSA, a second test was performed using the same panel of color-coded beads coated with recombinant HLA antigens, and a specific anti-human C3d conjugate (Lifecodes LSA C3d detection, Immucor, Stamford, CT, USA) according to manufacturer’s protocol. This assay determined whether the antibodies had the ability to bind complement components. For both identification assays, the interpretation was performed manually, using dedicated software (Match IT! Antibody version 1.1.0.2. Immucor, Stamford, CT, USA). For both standard and C3d LSA assays, the same cut-off value was used A specificity was considered as positive when background corrected mean fluorescence intensity (MFI) (BCM) for IgG/Raw value for C3d was higher than 500 and antigen density–background corrected ratio (AD-BCR) for IgG and BCR-Neg for C3d was higher than 5.

### 2.5. Immunologic Treatment

According to our protocol, all heart transplant patients underwent induction therapy with anti-thymocyte globulin (ATG). Baseline anti-rejection therapy following induction was a combination of T0 adjusted anticalcineurin treatments (Ciclosporin or Tacrolimus), mycophenolate mofetil, and low dose corticosteroids).

### 2.6. Statistical Analysis

Statistical analysis was performed using SPSS (version 27, IBM). Categorical variables were expressed as frequencies and percentages, and continuous variables as means and standard deviations, or median and interquartile range (IQR). The comparison of categorical variables was carried out with the chi-square test and the comparison of continuous variables with Student’s t test or Wilcoxon-Mann–Whitney test when necessary. The two-tailed significance level was set at *p* < 0.05.

Survival analysis using the Kaplan–Meier method and survival curves were compared by log-rank test. Association between DSA appearance and outcomes were assessed using Cox regression with time-dependent covariables in order to avoid immortal time bias [23]. Within this time-dependent model, the association between DSA and outcome was considered only after the occurrence of DSA (as no association between DSA and outcome can be causal before the development of DSA). To assess specific impact of complement-binding, a comparison between DSA positive C3d negative (DSA+C3d−) and DSA positive C3d positive (DSA+C3d+) outcome after DSA appearance was performed using Cox regression model and regression with time-dependent covariables with recipient sex, recipient age, CMV status at transplantation, number of cellular rejection episodes ≥ grade 2, total number of mismatches, creatinine at time of coronary angiography, and ejection fraction at time of coronary angiography.

The database was closed on 1 September 2020.

## 3. Results

### 3.1. Baseline Characteristics

The final population includes 282 patients. The analysis of DSA complement-binding ability showed the presence of non-complement-binding donor-specific anti-HLA antibodies in 22 patients (8%) and complement-binding donor-specific anti-HLA antibodies in 29 patients (10%). The mean follow-up time was 16.4 years after transplantation in the no DSA group and 14.3 and 15.5 in the DSA+C3d− and DSA+C3d+ groups, respectively. Anti-HLA antibodies were detected with a median time of 7.7 years post-transplant in the DSA+C3d− group and 10.1 years in the DSA+C3d+ group. Patients’ characteristics at time of transplantation were similar, including comparison for recipient and donor age, recipient gender, cold ischemia time, and creatinine at the time of angiography (Table 1).

### 3.2. Factors Associated with De Novo DSA

The number of A, B, DR, and DQ mismatches and the total number of mismatches were significantly higher in patients developing a donor specific antibody (0.7, 1.1, and 1.0 for the DSA−, DSA+C3d− et DSA+C3d+ groups, respectively) without significant difference in the number of rejection episodes ≥ grade 2, (*p* = 0.129) (Table 2).

### 3.3. Impact of C3d Binding Ability on Combined Criteria of Severe CAV or Death

Mean survival time free from severe CAV was 19.0, 19.0, and 13.9 years for DSA-, DSA+C3d-, and DSA+C3d+, respectively, *p* = 0.003 by the log-rank test (Figure 1). Presence of de novo DSA+ lacking C3d fixing ability (DSA+C3d−) was associated in univariable analysis with an increased 13% HR (1.13 (0.53–2.43) *p* = 0.745) on combined CAV or death. When C3d fixing ability was detected (DSA+C3d+) the HR for severe cardiac allograft vasculopathy or death was almost tripled (2.81 (1.69–4.67) *p* < 0.001). In multivariate analysis, HR for combined criteria of severe CAV or death was 1.04 (0.43–2.47) *p* = 0.937 and 4.31 (2.40–7.74) *p* < 0.001, respectively, for DSA+C3d− and DSA+C3d+ (Figure 2). Survival free from severe CAV after DSA appearance was shorter in DSA+C3d+ group versus DSA+C3d− (Figure 3). With DSA+C3d− group as reference, adjusted HR for severe CAV or mortality was 3.74 (1.31–10.65), *p* = 0.014 for DSA+C3d+ group.

### 3.4. Impact of DSA and C3d Binding Ability on Long-Term Mortality

Mean survival time was 21.9, 20.3, and 16.4 years for DSA−, DSA+C3d−, and DSA+C3d+, respectively, *p* = 0.027 by the log-rank test (Figure 4). Presence of de novo DSA+C3d− and DSA+C3d+ was associated in univariable analysis with a hazard ratio of 1.32 (0.61–2.83) *p* = 0.481 and 2.74 (1.67–4.49) *p* < 0.001, respectively of combined criteria of severe cardiac allograft vasculopathy or death. In multivariate analysis, HR for mortality was 1.08 (0.45–2.61) *p* = 0.866 and 3.48 (1.97–6.15) *p* < 0.001 for DSA+C3d− and DSA+C3d+, respectively (Figure 2). Survival after DSA appearance was shorter in DSA+C3d+ group versus DSA+C3d− (Figure 3). With DSA+C3d− group as reference, adjusted HR for mortality was 3.18 (1.13–8.99), *p* = 0.029 for DSA+C3d+ group.

## 4. Discussion

To our knowledge, we described the largest cohort of heart transplant recipients screened for the presence of complement-binding donor-specific anti-HLA antibodies. The long follow-up time and the use of time-dependent models accounting for the development of DSA during follow-up (i.e., taking into account all of the available follow-up) allowed us to highlight long-term predictors of mortality. The results of this study aim to show, for the first time, the deleterious impact of complement-binding donor-specific anti-HLA antibodies on a combined endpoint of cardiac allograft vasculopathy, and on long-term mortality, and on long-term mortality regardless of the diagnostic of AMR.

### 4.1. Strategy of DSA Follow-Up after Solid Organ Transplantation

Our observations plead for an annual screening of DSA even in absence of CAV or rejection. There are several methods for detecting circulatory anti-HLA antibodies before and after HT, they have been described in detail in a recent review [24]. Luminex assays seems more sensitive and specific compared with complement-dependent lymphocytotoxicity to predict the complement-binding capability of HLA IgG antibodies [25]. To our knowledge, there are no strong data supporting monitoring of a specific fraction of complement (i.e., C1q or C3d) and the choice should depend on test availability and team preference.

### 4.2. Impact of De Novo DSA on Risk of CAV and Mortality

Many studies have shown the risk of long term DSA appearance in the first years post transplantation [26,27]. One interesting point in our study is the long delay of DSA appearance and the detection in systematic follow-up regardless of a rejection. This late appearance could be explained by a reduction in immunosuppressive treatment, medication non-adherence, infection, other organs transplantation or blood transfusion [28,29].

In cardiac transplantation, whereas the effect of pretransplant DSA on long-term survival is still controversial [10,30], the deleterious effect of de novo DSA is more convincingly agreed [1,31,32,33]. To date, only few studies have evaluated the role of de novo complement-binding DSA in heart transplantation [20,34,35,36]. Zeevi et al. and Farrero Torres et al. found an association between complement-binding donor-specific anti-HLA antibodies and antibody mediated rejection [34,35]. In these studies, there was no difference in survival due to sample size and follow-up duration. In a pediatric population, Das et al. found C1q binding DSA (an equivalent of C3d binding abilities) as an independent risk for CAV with a hazard ratio of 3.25 [20]. C1q binding DSA were also associated with a risk of graft lost [20]. More recently, an association between C1q and CAV was found in adults [37]. In the last international consensus conference on management of antibodies in heart transplantation, only 35% of 103 participants from 75 centers routinely evaluated DSA complement fixing ability in DSA-positive patients [6]. This result could be explained by a limited availability of the technique, and doubts about the clinical benefits of complement-binding donor-specific anti-HLA antibodies detection.

We highlighted, for the first time, the clinical implication of de novo complement-binding donor-specific anti-HLA antibodies on risk of CAV and heart transplant survival regardless of the context of antibody mediated rejection.

As impact of de novo antibodies should be assessed only after their appearance and detection. We used a Cox regression model with time-dependent covariables. We have shown that the presence of complement-binding DSA was associated with a 4-fold risk of a combined criteria of severe CAV or death and more than 3-fold risk of death.

### 4.3. Implications for Clinical Practice

According to our opinion, there are three main implications for clinical practice.

First, we have shown that almost one in five patients will develop de novo DSA which is consistent with literature [27]. It justifies an annual assessment of DSA.

Second, complement-binding ability should be assessed systematically in presence of de novo DSA to stratify prognostic. As there is an increased risk of CAV, close systematic coronary monitoring is necessary in patients with complement-binding DSA.

Third, a recent study has shown the interest of complement inhibition to prevent AMR in immunologically high-risk heart allograft recipients [38]. Although there is no prospective study showing the benefit associated with complement inhibition on long-term mortality or CAV it could be reasonable to evaluate these treatments in future studies.

### 4.4. Study Limitations

Our study has several limitations. First, the observational retrospective design is subject to biases inherent in such studies. Second, we have not collected data on antibody mediated rejection, medication adherence and corticoid withdrawal. However, corticoid withdrawal is not widely used in our center and is limited to patients with steroid complications. Similarly, statin prescription and cholesterol levels were not assessed nor reported in this article, however our protocol requires all patients to be on pravastatin (40 mg od) unless not tolerated. Third, we included patients at the time of angiography and patients without at least one angiography were excluded. However, selecting all consecutive patients with coronary angiography limits bias selection and allow an assessment of long-term outcome associated with complement-binding DSA. The Kaplan–Meier curves in Figure 1 and Figure 3 represent the prognosis of patients based on antibody status even though DSAs appear after 8 to 10 years. However, as shown in the figures, the difference in prognosis appears after the appearance of antibodies. We used a Cox regression with time-dependent covariables to assess the impact of de novo antibodies only after appearance.

## 5. Conclusions

The pathogenicity of de novo DSA appears to depend on complement-binding ability. Complement-binding de novo DSA were associated with clinical outcome after heart transplantation whereas DSA without C3d-binding ability were not. A lifetime, annual monitoring of DSA and complement-binding ability seems mandatory.

## Figures and Tables

**Figure 1 jcm-11-03731-f001:**
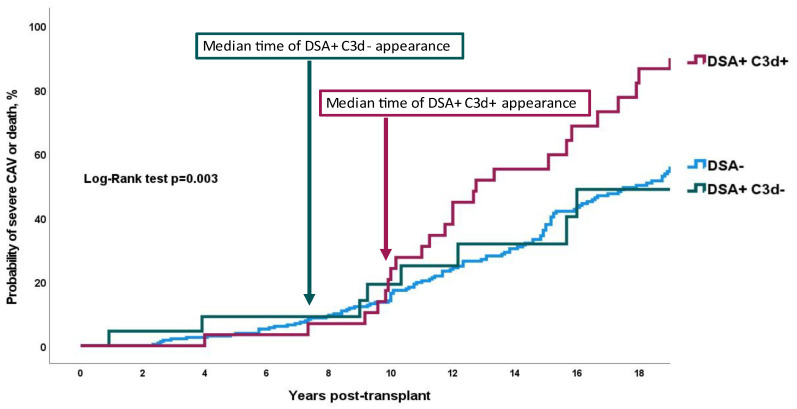
Probability of severe CAV or death according to DSA and C3d status. Log rank test *p* = 0.003. De novo C3d-binding DSA were associated with severe cardiac allograft vasculopathy and death. Non-complement-binding donor-specific anti-HLA antibodies were not associated with the risk of severe CAV or death. DSA = donor-specific anti-HLA antibodies; DSA+C3d+ = complement-binding donor-specific anti-HLA antibodies; DSA+C3d− = non-complement-binding donor-specific anti-HLA antibodies.

**Figure 2 jcm-11-03731-f002:**
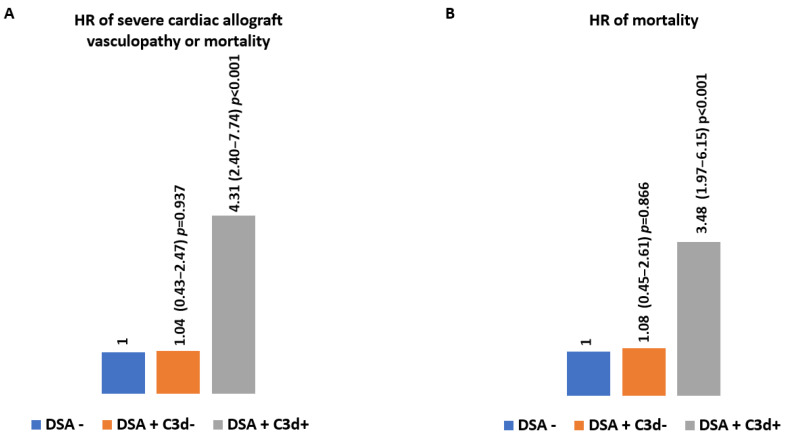
Association between outcomes and DSA and C3d status. Hazard ratio using time dependent Cox regression multivariate model with recipient sex, recipient age, total number of mismatches, number of rejection episodes ≥ grade 2, CMV mismatch, creatinine, and ejection fraction at time of coronary angiography. (**A**) severe cardiac allograft vasculopathy or mortality; (**B**) mortality.

**Figure 3 jcm-11-03731-f003:**
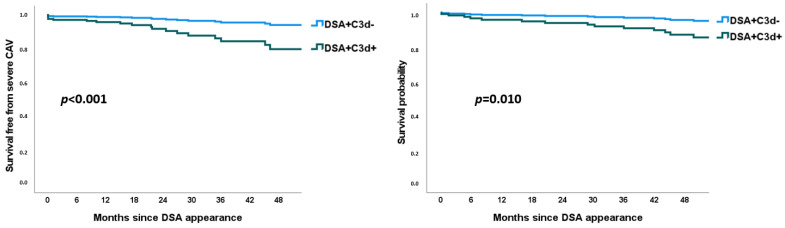
Adjusted Cox regression survival curves. Adjustment with recipient sex, recipient age, CMV status at transplantation, number of cellular rejection episodes ≥ grade 2, total number of mismatches, creatinine at time of coronary angiography, and ejection fraction at time of coronary angiography. CAV = cardiac allograft vasculopathy; DSA = donor specific antibodies.

**Figure 4 jcm-11-03731-f004:**
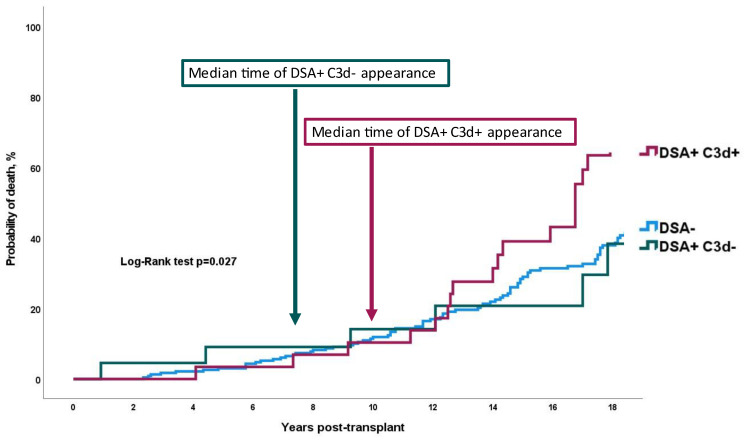
Probability of mortality according to DSA and C3d status. Log rank test *p* = 0.027. De novo C3d-binding DSA were associated with long-term mortality whereas non-complement-binding donor-specific anti-HLA antibodies were not. DSA = Donor-specific anti-HLA antibodies; DSA+C3d+ = complement-binding donor-specific anti-HLA antibodies; DSA+C3d− = non-complement-binding donor-specific anti-HLA antibodies.

**Table 1 jcm-11-03731-t001:** Recipients, donors, and surgical factors according to DSA status.

			DSA Status			*p*-Value
	Missing (%)	Total n = 282	NEG n = 231 (82)	DSA C3d− n = 22 (8)	DSA C3d+ n = 29 (10)	
	Recipient characteristics					
Follow-up time, years	0 (0)	16.2 (6.82)	16.4 (7.1)	14.3 (6.7)	15.5 (4.6)	0.479
Recipient age at transplantation, years	0 (0)	51 (41–58)	51 (41–59)	50 (32–56)	47 (37–57)	0.275
Male sex, n (%)	0 (0)	237 (84)	190 (82)	19 (86)	28 (97)	0.134
Ejection fraction, %	0 (0)	60 (55–67)	60 (55–67)	60 (55–68)	60 (52–65)	0.750
Corticoid regimen, n (%)	0 (0)	177 (63)	142 (62)	16 (73)	19 (66)	0.551
Calcineurin inhibitors, n (%)	0 (0)	270 (96)	221(96)	22 (100)	27 (93)	0.478
mTOR inhibitors, n (%)	0 (0)	40 (14)	32 (14)	5 (23)	3 (10)	0.429
Purine inhibitors, n (%)	0 (0)	214 (76)	177 (77)	15 (68)	22 (76)	0.676
Creatinine, μmol/L	36 (13)	116 (92–143)	115 (93–143)	100 (81–150)	119 (99–146)	0.673
CMV positive status at time of transplantation	0 (0)	135 (48)	112 (48)	11 (50)	12 (41)	0.301
	Donors and surgical factors					
Donor age, years	16 (6)	35 (24–44)	34 (23–44)	39 (33–48)	35 (25–43)	0.144
Cytomegalovirus mismatch, n (%)	17 (6)	46 (17)	32 (15)	8 (36)	6 (21)	0.034
Sex mismatch, n (%)	15 (5)	30 (11)	23 (11)	4 (18)	3 (11)	0.560
Cold ischemia time, minutes	11 (4)	201 (175–247)	201 (175–251)	212 (170–233)	191 (175–245)	0.926

CMV = Cytomegalovirus.

**Table 2 jcm-11-03731-t002:** Immunological factors according to DSA status.

	DSA Status			
	NEG n= 231	DSA C3d− n = 22 (7.6%)	DSA C3d+ n = 29 (10.0%)	*p* Value
Number of mismatches A	1.1 (0.9)	1.7 (0.6)	1.3 (0.8)	0.010
Number of mismatches B	1.2 (0.9)	1.6 (0.6)	1.6 (0.8)	0.039
Number of mismatches DR	1.0 (0.9)	1.5 (0.8)	1.4 (0.9)	0.009
Number of mismatches DQ	0.8 (0.8)	1.4 (0.8)	1.0 (0.8)	0.017
Number of HLA mismatches	4.2 (3.0)	6.1 (2.0)	5.2 (2.7)	0.004
Number of cellular rejection episodes ≥ grade 2	0.7 (1.4)	1.1 (1.4)	1.0 (2.6)	0.129
Delay of DSA appearance from HT (years)	-	7.7 (3.1–13.3)	10.1 (7.7–11.7)	-
Cumulative MFI for DSA	-	4400 (1775–7875)	17,000 (14,050–31,250)	-
Pic MFI for DSA	-	4000 (1775–6975)	15,900 (12,350–16,950)	-
Cumulative MFI for C3d	-	-	8300 (4350–13650)	-

DSA, donor-specific anti-HLA antibodies; MFI = Mean fluorescence intensity. Values are expressed as median (interquartile 25–75) or mean (standard deviation).

## Data Availability

The study did not report any data.

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
