# Peer review of "De Novo Complement-Binding Anti-HLA Antibodies in Heart Transplanted Patients Is Associated with Severe Cardiac Allograft Vasculopathy and Poor Long-Term Survival"

_jcm, 2022, doi:10.3390/jcm11133731_

Round 1

Reviewer 1 Report

Summary

The presence of serum donor specific anti-HLA antibody (DSA) contributes to the development of antibody mediated rejection, endothelial dysfunction, and graft failure in solid organ transplantation. The incidence of de novo DSA in the circulation systems ranges around 10-40%  in solid organ transplantation (McCaughan & Tinckam, 2018). Presence of de novo DSA after heart transplantation is reported to increasing risks of rejection and allograft vasculopathy (CAV) through activation of complement systems (Bouquegneau et al., 2018; McCaughan & Tinckam, 2018).  

The present study submitted by G. Baudry, et al focus on the serum de novo DSA with or without the complement (C3q) binding. The endpoints include development of CAV (scores ≧ 2) and long-term survival. The research is not novel, but provide important insights to the clinical impact of the development of circulating DSA after heart transplantation. A few points that need to be addressed are as follows:

Major--

Results:

1.      What are the prevalence of antibody mediated rejection (AMR)  in this cohort? Are there differences in prevalence of AMR among these subgroups?

Discussion:

1.      Would the authors describe their management protocol with a brief literature review on the detection of complement activating DSA in serum.

Minor--

1.      Table 2. What do the number in the round brackets mean for variables of "Number of mismatches A, B, DR, etc." ? Please indicate in the footnotes.

2.    The Figures 1, 2 and 3 were lost in the manuscript for review. 

Author Response

The presence of serum donor specific anti-HLA antibody (DSA) contributes to the development of antibody mediated rejection, endothelial dysfunction, and graft failure in solid organ transplantation. The incidence of de novo DSA in the circulation systems ranges around 10-40%  in solid organ transplantation (McCaughan & Tinckam, 2018). Presence of de novo DSA after heart transplantation is reported to increasing risks of rejection and allograft vasculopathy (CAV) through activation of complement systems (Bouquegneau et al., 2018; McCaughan & Tinckam, 2018). 

The present study submitted by G. Baudry, et al focus on the serum de novo DSA with or without the complement (C3q) binding. The endpoints include development of CAV (scores ≧ 2) and long-term survival. The research is not novel, but provide important insights to the clinical impact of the development of circulating DSA after heart transplantation. A few points that need to be addressed are as follows

Major--

Results:

  1. What are the prevalence of antibody mediated rejection (AMR) in this cohort? Are there differences in prevalence of AMR among these subgroups?

We agree that AMR prevalence is an important information. However, our registry contains only information on cellular rejection.

We added this sentence in limitation paragraph: “Second, we haven’t collected data on antibody mediated rejection, medication adherence and corticoid withdrawal.”

Discussion:

  1. Would the authors describe their management protocol with a brief literature review on the detection of complement activating DSA in serum.

We thank the reviewer for this advice. We edited the methods paragraph as follow:

HLA antibody screening during follow up.

Post-transplantation monitoring for DSA were performed at 1, 3, 6, and 12 months post-operatively according to ISHLT guidelines (6). Beyond one year post transplant, patients without DSA were monitored annually or more frequently in case of  sensitization.

In the discussion paragraph we added :

… This result could be explained by a limited availability of the technique and doubts about clinical benefice of complement-binding donor-specific anti-HLA antibodies detection.  There are several methods for detecting circulatory anti-HLA antibodies before and after HT, they have been described in detail in a recent review (24).  Luminex assays seems more sensitive and specific compared with complement-dependent lymphocytotoxicity to predict the complement-binding capability of HLA IgG antibodies (25). To our knowledge, there are no strong data supporting monitoring of a specific  fraction of complement (ie C1q or C3d) and the choice should depend on test availability and team preference.”

Minor--

  1. Table 2. What do the number in the round brackets mean for variables of "Number of mismatches A, B, DR, etc." ? Please indicate in the footnotes.

We apologize for this omission. We thank the reviewer for this remark and have corrected it in the revised manuscript.

  1. The Figures 1, 2 and 3 were lost in the manuscript for review.

We are sorry to hear that have made sure the figures are part of the revised manuscript. 

Reviewer 2 Report

The Authors present this interesting article with the title :"De-novo complement binding anti-HLA antibodies in heart transplanted patients is associated with severe cardiac allograft vasculopathy and poor long-term survival. The primary outcome was a composite criterion of severe CAV or mortality. As the impact of de novo antibodies should be assessed only after appearance, we used a Cox regression with time-dependent covariables. The author concluded that Donor Specific Antibodies were associated with worse clinical outcome when binding complement. DSA and their complement binding ability should thus be screened for to optimize heart transplant patient follow-up. 

1) for the high scientific impact of the paper and content, 2)the manuscript is really well written with 3)good languages and 4)graphic presentation. Thank you and congratulation.

Ignazio Condello

Author Response

The Authors present this interesting article with the title :"De-novo complement binding anti-HLA antibodies in heart transplanted patients is associated with severe cardiac allograft vasculopathy and poor long-term survival. The primary outcome was a composite criterion of severe CAV or mortality. As the impact of de novo antibodies should be assessed only after appearance, we used a Cox regression with time-dependent covariables. The author concluded that Donor Specific Antibodies were associated with worse clinical outcome when binding complement. DSA and their complement binding ability should thus be screened for to optimize heart transplant patient follow-up.

1) for the high scientific impact of the paper and content, 2)the manuscript is really well written with 3)good languages and 4)graphic presentation. Thank you and congratulation.

Ignazio Condello

We thank Dr Condello for his kind review.